# Energetic instability of passive states in thermodynamics

Carlo Sparaciari [1], David Jennings[2,3] & Jonathan Oppenheim[1]

Passivity is a fundamental concept in thermodynamics that demands a quantum system's energy cannot be lowered by any reversible, unitary process acting on the system. In the limit of many such systems, passivity leads in turn to the concept of complete passivity, thermal states and the emergence of a thermodynamic temperature. Here we only consider a single system and show that every passive state except the thermal state is unstable under a weaker form of reversibility. Indeed, we show that given a single copy of any athermal quantum state, an optimal amount of energy can be extracted from it when we utilise a machine that operates in a reversible cycle. This means that for individual systems, the only form of passivity that is stable under general reversible processes is complete passivity, and thus provides a physically motivated identification of thermal states when we are not operating in the thermodynamic limit.

[1] Department of Physics and Astronomy, University College London, London WC1E 6BT, UK. [2] Department of Physics, University of Oxford, Oxford OX1 3PU, UK. [3] Department of Physics, Imperial College London, London SW7 2AZ, UK. Correspondence and requests for materials should be addressed to C.S. (email: carlo.sparaciari.14@ucl.ac.uk) or to D.J. (email: david.jennings@physics.ox.ac.uk) or to J.O. (email: j.oppenheim@ucl.ac.uk)

Within thermodynamics, heat engines are devices that operate in a thermal context so as to extract ordered energy in the form of work. The canonical scenario involves an engine that operates cyclically between two temperatures $T_{hot}$, $T_{cold}$ and performs a quantity of mechanical work. To do so, the engine absorbs heat from the hot reservoir, converts some of this energy to mechanical work and releases heat into the cold reservoir in accordance with the second law of thermodynamics. The largest possible efficiency, $\eta = 1 - \frac{T_{cold}}{T_{hot}}$, occurs for the reversible Carnot engine[1, 2] and provides a fundamental thermodynamic bound on the amount of ordered energy that can be obtained. Carnot engines, and more in general heat engines, have been extensively studied in the microscopic regime[3–28] (as well as, of course, in the macroscopic regime).

However, the issue of ordered energy extraction can also be considered in scenarios in which no notion of temperature exists, and can provide a broader notion of equilibrium states. For example, more general equilibrium states can occur in physical realisations when a system has been perturbed and has not had enough time to fully thermalise. They can also arise in the context of non-equilibrium steady states[29], and even when we consider closed systems in a pure state, but we only have access to a small portion of the system. Given a quantum system in a state $\rho$ one can ask if it is possible to extract energy from it solely by performing a reversible unitary transformation on the system. The largest amount of ordered energy that was previously thought to be extractable (the 'ergotropy', see refs.[30–32]) depends nontrivially on the quantum state. If no energy can be extracted in this way, then $\rho$ is called passive[33–37] and constitutes a primitive form of equilibrium.

In this work, we consider a scenario that is intermediate between heat engines and passive states, and is motivated by the fact that a work extraction machine should be considered as a system, which is involved in the process. Our core question is whether there exist passive states $\rho_S$ for which energy can be extracted if one performs a reversible unitary process over the system S together with a second quantum system M, which starts and finishes in the same quantum state $\rho_M$. This second quantum system is the machine which, in analogy with the working body of a Carnot cycle, undergoes a cyclic evolution. This class of processes (which have been termed catalytic thermal operations[38]) is reminiscent of the ones taking place inside heat engines. However, there are some important differences between heat engines and our machine, not last the fact that our scheme does not involve actual thermal reservoirs, and there are not physical temperatures involved.

Rephrasing our core question (namely, whether it is possible to reversibly reduce the energy of a single passive state) in terms of optimal work extraction performed by a thermal machine appears to be rather convenient. In this setting, the main system S, described by the passive state, plays the role of both thermal reservoirs of an heat engine, while the additional system M can be seen as the machine, which exchanges energy between these two reservoirs in a cyclic manner. It is worth noting that, while the above identification of a passive state in terms of two thermal reservoirs is convenient for analysing our protocol, one ought to be cautious about assigning a physical meaning to it, due to the obvious differences between single systems in a passive state and thermal reservoirs.

Our study on passivity extends the set of allowed operations (which is originally composed solely of unitary operations) with the possibility of interacting with an additional system, which undergoes a cyclic dynamics. Extending the set of operations to this broader class seems reasonable, especially in light of the fact that microscopic machines can nowadays be realised in the laboratory[4, 7, 11, 12, 39, 40].

In this paper, we show that energy can always be extracted from single copies of athermal passive states by means of the above set of allowed operations. We provide an explicit protocol for the energy extraction, involving an ancillary system whose local state is left invariant by the evolution. We show that, in general, correlations between the main system and the ancilla are created during the extraction protocol. However, when the dimension of the ancillary system goes to infinity, these correlations can be reduced to an infinitesimal amount, which results in a reversible (energy-extracting) dynamics on the main system. Crucially, this result has fundamental implications for the notion of passivity. In fact, if energy can be extracted from a passive state with these reversible processes, and no entropy is generated, then it seems that associating passivity of the state is a restricted idealisation, unstable under this simple extension. Thus, our work provide a way to single out the thermal state, and consequently to recover a notion of temperature, without having to take the thermodynamic limit, or to consider thermalisation scenarios. Within the passivity setting, the thermal state was previously identified as the only passive state from which no work could be extracted in the thermodynamic limit. Indeed, all other passive states become active in this limit, that is, work can be extracted from many copies of them with a global unitary operation[31]. However, since a theory of thermodynamics can also be formulated without taking the thermodynamic limit, one should be able to single out the thermal state even when considering single systems, and the results of our study show how this is possible.

## Results

**Passive states.** Consider a finite-dimensional quantum system associated with the Hilbert space $\mathcal{H} \equiv \mathbb{C}^d$ (a qudit), with Hamiltonian $H = \sum_{i=0}^{d-1} E_i |i\rangle\langle i|$, and described by the state $\rho$. We say that the state $\rho$ is passive iff its average energy cannot be lowered by acting on it with unitary operations, that is,

$$\text{Tr}[H\rho] \leq \text{Tr}[HU\rho U^\dagger], \forall U \in \mathcal{B}(\mathcal{H}), UU^\dagger = U^\dagger U = \mathbb{I}. \quad (1)$$

This implies that no work can be extracted from the state via a unitary process, since by conservation of energy, lowering the energy of a system would mean that this energy has been transferred to a work storage device.

We can also introduce a more restrictive notion of passivity. Let us consider $n \in \mathbb{N}$ independent and identically distributed (i.i.d.) copies of our system, with a total Hamiltonian $H^{(n)} = \sum_{i=1}^{d} H_i$, where each $H_i$ is a single system Hamiltonian acting on a different copy of the system. The state of this global system is described by $\rho^{\otimes n}$. Then, we say that the state $\rho$ is completely passive if and only if the state $\rho^{\otimes n}$ is passive for all $n \in \mathbb{N}$, while the state $\rho$ is k-activable if $\rho^{\otimes n}$ becomes active for $n \geq k$. It can be shown[33] that the completely passive states of a system with Hamiltonian $H$ are the ones satisfying the KMS condition[41–43]. Specifically, these states are the ground state and the thermal states with temperature $\beta \geq 0$, that is, $\tau_\beta = e^{-\beta H}/Z$ with $Z = \text{Tr}[e^{-\beta H}]$. Any state which is not of this form, is called athermal.

A characterisation of all passive states can be easily obtained. A system in a passive state is such that the ground state has the highest probability of being occupied, and the probability of occupation decreases as the energy associated with the eigenstate of $H$ increases, Fig. 1. Specifically, a state $\rho$ is passive iff $\rho = f(H)$, where $f$ is a monotone non-increasing function. Simply put, this means that the state can expressed as

$$\rho = \sum_{i=0}^{d-1} p_i |i\rangle\langle i|, \text{ such that } p_i \geq p_{i+1} \forall i = 0, \dots, d-2, \quad (2)$$

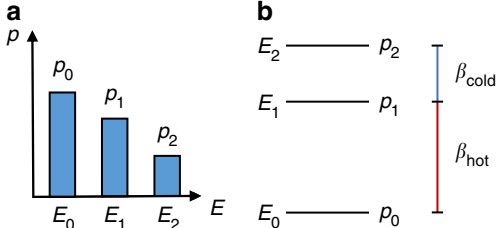

**Fig. 1** Passive state spectrum and virtual temperatures. **a** The spectrum of a qutrit passive state $\rho = \sum_{i=0}^{2} p_i |i\rangle\langle i|$ over the eigenbasis of its Hamiltonian $H = \sum_{i=0}^{2} E_i |i\rangle\langle i|$. The occupation probabilities are ordered in a decreasing order, from the one associated with the ground state of $H$ to the one associated with the maximally excited one, as per definition in Eq. (2). **b** A passive state can equally be described by virtual temperatures. Indeed, for each pair of eigenvalues of $\rho$, say $p_i$ and $p_j$, we can define a virtual temperature $\beta_{ij}$ through the relation $p_i/p_j = e^{-\beta_{ij}(E_i - E_j)}$, where $E_i$ ($E_j$) is the energy level associated with the eigenstate $|i\rangle$ ($|j\rangle$). In the figure, the pair of eigenstates $|0\rangle$ and $|1\rangle$ is associated with the hot temperature $\beta_{\text{hot}}^{-1}$, while the pair $|1\rangle$ and $|2\rangle$ is associated with the cold temperature $\beta_{\text{cold}}^{-1}$. The temperature associated with $|0\rangle$ and $|2\rangle$ is an average of the other two temperatures

where $\{|i\rangle\}_{i=0}^{d-1}$ are the eigenvectors of $H$, ordered so that $E_i \le E_{i+1}$ for all $i$ (for the case of equal energies $E_i = E_{i+1}$ we must make an additional stability assumption to ensure that $p_i = p_{i+1}$).

We can describe the probability distribution of the passive state $\rho$ by using virtual temperatures[15, 44]. In fact, for any given passive state, we can associate a (non-negative) virtual temperature with each pair of its eigenstates. For example, if we consider the pair $(|i\rangle, \langle j|)$, we define the virtual temperature associated with them as the $\beta_{ij}^{-1} \ge 0$ such that

$$\frac{p_i}{p_j} =: e^{-\beta_{ij}\left(E_i - E_j\right)}, \tag{3}$$

where $p_i$ is the probability of occupation of the state $|i\rangle$, and $E_i$ is the energy associated with the state (similarly for $j$). Thus, each pair of states can be regarded as an effective thermal state at a specific temperature. When all pairs of states has the same virtual temperature, we have that the passive state is completely passive, that is, it is the thermal state of $H$ at that temperature.

**The core protocol**. We now introduce a device that extracts work by acting individually on a passive state. The device is composed by two elements, namely, a main system in a specific passive state, and a qudit "machine" system. As we will see, the following protocol is independent of the Hamiltonian of the machine, and we set it to be the trivial Hamiltonian $H = \mathbb{I}$ for simplicity. Furthermore, we consider the main system to be a qutrit, since the protocol can be easily generalised to quantum $d$-level systems. The Hamiltonian of the main system is

$$H_{\text{P}} = \sum_{i=0}^{2} E_i |i\rangle\langle i|_{\text{P}}, \tag{4}$$

where $E_i \le E_{i+1}$, and it is described by the state

$$\rho_{\text{P}} = \sum_{i=0}^{2} p_i |i\rangle\langle i|_{\text{P}}, \tag{5}$$

where $p_i \ge p_{i+1}$ (Fig. 1).

In the following we parametrise the passive state $\rho_{\text{P}}$ in terms of virtual temperatures. We utilise such parametrisation because it allows us to draw a conceptual link between the current scenario

and the one of heat engines. This link turns out to be very convenient for the exposition of our protocol, but one should not assign to it any physical meaning; in fact, our scenario is completely different from the one in which heat engines operates. We are here considering single systems in a passive state, no physical temperatures are involved, and no actual thermal reservoirs are present.

The passive state is parametrised as follow; the virtual temperature $T_{\text{hot}} = \beta_{\text{hot}}^{-1} > 0$ is associated with the pair of eigenstates $(|0\rangle_{\text{P}}, |1\rangle_{\text{P}})$, and the virtual temperature $T_{\text{cold}} = \beta_{\text{cold}}^{-1} > 0$ is associated with the pair $(|1\rangle_{\text{P}}, |2\rangle_{\text{P}})$. We assume for simplicity that $T_{\text{hot}} > T_{\text{cold}}$, but a similar analysis applies for $T_{\text{hot}} < T_{\text{cold}}$. In Supplementary Note 1 the cycle is presented in full detail. The relation between the probability distribution of $\rho_P$ and the temperatures $T_{\text{hot}}$ and $T_{\text{cold}}$ is given by

$$\frac{p_1}{p_0} =: e^{-\beta_{\text{hot}}\Delta E_{10}}, \tag{6}$$

$$\frac{p_2}{p_1} =: e^{-\beta_{\text{cold}}\Delta E_{21}}, \tag{7}$$

where $\Delta E_{10} = E_1 - E_0 \ge 0$, and $\Delta E_{21} = E_2 - E_1 \ge 0$. Thus, by referring again to the conceptual link with heat engines, the pair of states $(|0\rangle_{\text{P}}, |1\rangle_{\text{P}})$ can be visualised as if it was a hot reservoir, while the pair of states $(|1\rangle_{\text{P}}, |2\rangle_{\text{P}})$ can be visualised as if it was a cold reservoir. It is worth noting that the other pair of states, $(|0\rangle_{\text{P}}, |2\rangle_{\text{P}})$, is associated with a virtual temperature that is intermediate between $T_{\text{cold}}$ and $T_{\text{hot}}$, as we can easily verify from Eqs. (6) and (7).

The protocol extracts work by means of the following cycle. A single system, described by the state $\rho_{\text{P}}$, is put in contact with the machine, described by the state $\rho_{\text{M}} = \sum_{j=0}^{d-1} q_j |j\rangle\langle j|_{\text{M}}$. Then, we perform $m$ swaps (we refer to them as hot swaps) between the hot "virtual" reservoir of the passive state and $m$ different pairs of states of $\rho_{\text{M}}$, followed by $n$ swaps (cold swaps) between the cold "virtual" reservoir and other $n$ different pairs of states of $\rho_{\text{M}}$. In order to perform the swaps on different pairs of states, we need the machine to have at least $m + n$ levels, and therefore we fix $d = m + n$. Specifically, we apply the following unitary operation to the global system

$$S_{m,n} = S_{(1,2)}^{(0,m)} \circ S_{(1,2)}^{(m,m+1)} \circ S_{(1,2)}^{(m+1,m+2)} \circ \dots$$
$$\circ S_{(1,2)}^{(m+n-2,m+n-1)} \circ S_{(0,1)}^{(m-1,m+n-1)} \tag{8}$$
$$\circ S_{(0,1)}^{(m-2,m-1)} \circ S_{(0,1)}^{(m-3,m-2)} \circ \dots \circ S_{(0,1)}^{(0,1)},$$

where the operator $S_{(a,b)}^{(c,d)}$ is a swap between system and machine, performed through the permutation $|a\rangle_P |d\rangle_{\text{M}} \leftrightarrow |b\rangle_P |c\rangle_{\text{M}}$. A graphical representation of this global operation is shown in Fig. 2, where each swap is depicted by an arrow acting over the states of the machine. Although in the figure we represent the eigenstates of $\rho_{\text{M}}$ in a ladder, they are all associated with the same energy, and therefore the order in which we present them is only functional for visualising the cycle $S_{m,n}$.

In order for the protocol to be cyclic, we need the local state of the machine to end up in its initial state. Therefore, we impose the following constraint on the state of the machine,

$$\rho_{\text{M}} \overset{!}{=} \text{Tr}_{\text{P}}\left[ S_{m,n}(\rho_{\text{P}} \otimes \rho_{\text{M}}) S_{m,n}^{\dagger} \right]. \tag{9}$$

Through Eq. (9) we can express the probability distribution of the machine in terms of the passive state $\rho_P$. In our model, we do not explicitly include an additional system (a battery) for storing the energy we extract from the passive state. Instead, we implicitly

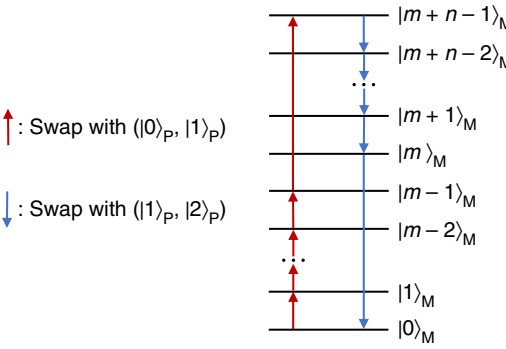

Fig. 2 The action of the cycle over the machine. The cycle $S_{m,n}$ is represented in a pictorial way over the eigenstates of the $d$-dimensional machine (where $d = m + n$). Notice that the machine has a trivial Hamiltonian, and we order the eigenstates to only simplify the visualisation of the cycle. The upward arrow connecting two eigenstates of the machine represents a swap between these two states and the pair $(|0\rangle_P, |1\rangle_P)$ of the passive state. The downward arrow connecting two eigenstates of the machine represents a swap between this pair and the pair $(|1\rangle_P, |2\rangle_P)$ of the passive state. We initially perform $m - 1$ swaps between $(|0\rangle_P, |1\rangle_P)$ and $\{(|j\rangle_M, |j + 1\rangle_M)\}_{j=0}^{m-2}$, and one swap between $(|0\rangle_P, |1\rangle_P)$ and $(|m - 1\rangle_M, |m + n - 1\rangle_M)$. Then, we perform $n - 1$ swaps between $(|1\rangle_P, |2\rangle_P)$ and $\{(|j\rangle_M, |j + 1\rangle_M)\}_{j=m}^{m+n-2}$, and one swap between $(|1\rangle_P, |2\rangle_P)$ and $(|0\rangle_M, |m\rangle_M)$. If we consider the arrow representation of swaps, we can see that the cycle is close, and this allows us to recover the local state of the machine M while extracting work

assume the existence of this work storage system, and we define the work extracted, $\Delta W$, as the difference in average energy between the initial and final state of the main system (as the machine $M$ has a trivial Hamiltonian, and no interaction terms are present between system and machine). Thus, we have that

$$\Delta W = \mathrm{Tr}[H_P(\rho_P - \tilde{\rho}_P)], \qquad (10)$$

where the final state of the system is

$$\tilde{\rho}_P = \mathrm{Tr}_M\left[S_{m,n}(\rho_P \otimes \rho_M)S_{m,n}^\dagger\right]. \qquad (11)$$

It is worth noting that the final state of system and machine will in general develop correlations. These correlations are classical, and without them work would not be extracted during the cycle. However, they do not compromise the re-usability of the machine if applied to another uncorrelated quantum system.

For a given system Hamiltonian $H_P$ and a given cycle $S_{m,n}$, we can investigate the amount of work we extract from the state $\rho_P$. In Supplementary Note 1 we provide all the necessary steps to evaluate $\Delta W$ in terms of the probability distribution of $\rho_P$. We can express this quantity as

$$\Delta W = \alpha(m\Delta E_{10} - n\Delta E_{21})\left(e^{\beta_{\mathrm{cold}} n\Delta E_{21}} - e^{\beta_{\mathrm{hot}} m\Delta E_{10}}\right), \qquad (12)$$

where $\alpha$ is a positive coefficient depending non-trivially on the probability distribution of $\rho_P$. For the class of passive states, we are considering (namely, the one in which $\beta_{\mathrm{cold}} > \beta_{\mathrm{hot}}$), we find that work can be extracted ($\Delta W > 0$) iff

1. The Hamiltonian $H_P$ is such that $m\Delta E_{10} > n\Delta E_{21}$.
2. The temperatures of the two virtual reservoirs are such that $\beta_{\mathrm{cold}} > \frac{m\Delta E_{10}}{n\Delta E_{21}}\beta_{\mathrm{hot}}$.

Thus, for a fixed cycle (defined by the parameters $m$ and $n$), and for a fixed Hamiltonian $H_P$, we find that work can only be

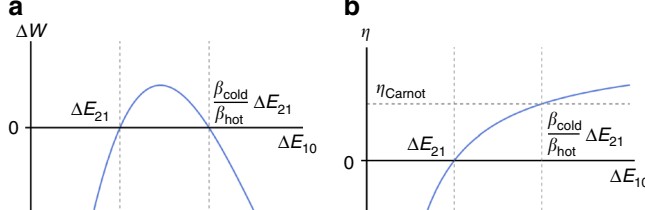

Fig. 3 Energy extraction and efficiency with a qubit machine. Consider a qutrit system with Hamiltonian $H_P = \sum_{i=0}^2 E_i |i\rangle\langle i|_P$, in the passive state $\rho_P = \sum_{i=0}^2 p_i |i\rangle\langle i|_P$. The simplest energy-extracting protocol involves a qubit machine, and energy is extracted from the passive state by performing a single hot and cold swap between system and machine. This cycle is analogous to the one studied in ref. [13]. **a** The energy extracted from a given passive state depends on the Hamiltonian of the system. This energy is positive iff the hot energy gap $\Delta E_{10} = E_1 - E_0$ lies inside the range $\left(\Delta E_{21}; \frac{\beta_{\mathrm{cold}}}{\beta_{\mathrm{hot}}} \Delta E_{21}\right)$, where $\Delta E_{21} = E_2 - E_1$ is the cold energy gap, and $\beta_{\mathrm{hot}}^{-1}$ ($\beta_{\mathrm{cold}}^{-1}$) is the virtual temperature associated with the pair of eigenstates $|0\rangle_P$ and $|1\rangle_P$ ($|1\rangle_P$ and $|2\rangle_P$). The above range is obtained from conditions 1 and 2. **b** The efficiency of the protocol $\eta$ is defined as the ratio between the energy extracted at the end of the protocol and the energy extracted when performing hot swap (that is, the total positive energy extracted during the cycle). When the hot energy gap $\Delta E_{10}$ lies in the correct range, the efficiency takes values between 0 and the Carnot efficiency

extracted if the virtual temperature $T_{\mathrm{cold}}$ is lower than $T_{\mathrm{hot}}$ by a multiplicative factor, which depends on the energy gaps of the Hamiltonian, see Fig. 3 for an example. In the next section we will show that, for a given Hamiltonian $H_P$, work can be extracted from any passive (but not completely passive) state, and we characterise the cycle which allows for this extraction.

If we analyse in a more detailed way the cycle, we find that the same amount of energy is gained during each swap between the machine $M$ and the hot virtual reservoir, that is

$$q_{\mathrm{hot}} = \alpha\Delta E_{10}\left(e^{\beta_{\mathrm{cold}} n\Delta E_{21}} - e^{\beta_{\mathrm{hot}} m\Delta E_{10}}\right), \qquad (13)$$

where $\alpha$ is the same positive coefficient of Eq. (12). Moreover, the same amount of energy is spent during each swap between the machine $M$ and the cold virtual reservoir,

$$q_{\mathrm{cold}} = \alpha\Delta E_{21}\left(e^{\beta_{\mathrm{cold}} n\Delta E_{21}} - e^{\beta_{\mathrm{hot}} m\Delta E_{10}}\right). \qquad (14)$$

Knowing the amount of energy exchanged during each swap allows us to evaluate the heat exchanged with the virtual reservoirs. In fact, if we identify the pair of levels $(|0\rangle_P, |1\rangle_P)$ with the hot virtual reservoir, then the energy exchanged during a swap with these levels can be considered as heat coming from the hot virtual reservoir. In this way, the total heat absorbed by the machine is

$$Q_{\mathrm{hot}} = m\, q_{\mathrm{hot}}, \qquad (15)$$

while the total heat provided to the cold virtual reservoir is

$$Q_{\mathrm{cold}} = n\, q_{\mathrm{cold}}. \qquad (16)$$

From Eqs. (15) and (16), we obtain that the work extracted can be expressed as $\Delta W = Q_{\mathrm{hot}} - Q_{\mathrm{cold}}$, as in a standard heat engine exchanging energy between two reservoirs. Once $Q_{\mathrm{hot}}$ and $Q_{\mathrm{cold}}$

are defined, we can evaluate an efficiency of this cycle, that is

$$\eta = \frac{\Delta W}{Q_{\text{hot}}} = 1 - \frac{n \Delta E_{21}}{m \Delta E_{10}}. \tag{17}$$

The efficiency of our protocol (when the machine is finite-dimensional) is sub-Carnot in the virtual temperatures, see Fig. 3. In fact, work can only be extracted when conditions 1 and 2 are satisfied, and these conditions implied $0 < \eta < 1 - \frac{T_{\text{cold}}}{T_{\text{hot}}}$. When we consider the case of an infinite-dimensional machine, we find that by a judicious choice of parameters we may obtain Carnot efficiency.

Once the cycle $S_{m,n}$ is ended, the local state of the main system is moved to a less energetic state. By solving Eq. (9), we find that the final state of the main system $\tilde{\rho}_P$ has the following probability distribution

$$p'_0 = p_0 + m \Delta P, \tag{18}$$

$$p'_1 = p_1 - (m + n) \Delta P, \tag{19}$$

$$p'_2 = p_2 + n \Delta P, \tag{20}$$

where the unit of probability $\Delta P$ depends on the initial state $\rho_P$, and it is given by

$$\Delta P = \alpha \left( e^{\beta_{\text{cold}} n \Delta E_{21}} - e^{\beta_{\text{hot}} m \Delta E_{10}} \right), \tag{21}$$

with $\alpha$ the same positive coefficient of Eq. (12). Due to condition 2, the unit $\Delta P > 0$, so that the probability of occupation of $|1\rangle_P$ is reduced in favour of the probabilities $p_0$ and $p_1$. Thus, energy is extracted from the passive state when $m \Delta P$ is moved from $p_1$ to $p_0$ (during the hot swaps), and part of this energy is used to move the probability $n \Delta P$ from $p_1$ to $p_2$ (during the cold swaps), see Supplementary Fig. 1.

**Work extraction from any passive state**. We now show that, for a given Hamiltonian $H_P$, work can be extracted from any passive but not completely passive state. In particular, we first show this for qutrit passive states, and we then generalise to the qudit case. Work extraction is achieved with the cycle presented in previous section, for specific values of the parameters $m$ and $n$. In what follows, we represent the passive state with the probabilities of occupation $\{p_0, p_1, p_2\}$, as opposed to the previous case in which the virtual temperatures were used. In this way, we can consider all possible scenarios, and we are not limited to the case in which a specific pair of eigenstates has a colder (hotter) virtual temperature than the other pair.

The Hamiltonian of the system $H_P$ is defined in Eq. (4), where the energy gap between ground and first excited state is $\Delta E_{10}$, and the gap between first and second excited states is $\Delta E_{21}$. We assume that

$$\exists M, N \in \mathbb{N} \text{ such that } M \Delta E_{10} - N \Delta E_{21} = 0, \tag{22}$$

that is, we ask the ratio between the two energy gaps to be rational. Notice that, even if the ratio is irrational, we can find a suitable $N$ and $M$ such that the condition is approximately satisfied. Once the relation between energy gaps is defined, we can

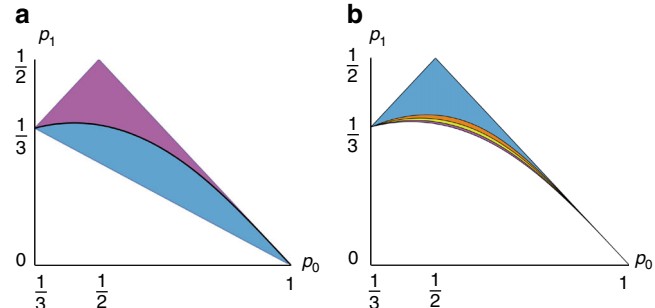

**Fig. 4** The region of qutrit passive states and the activable set. Consider a qutrit system with Hamiltonian $H_P = \sum_{i=0}^{2} E_i |i\rangle \langle i|_P$, whose state $\rho_P = \sum_{i=0}^{2} p_i |i\rangle \langle i|_P$ is passive. The Hamiltonian is fixed by choosing two $N, M \in \mathbb{N}$ such that $M \Delta E_{10} - N \Delta E_{21} = 0$, where $\Delta E_{10} = E_1 - E_0$ and $\Delta E_{21} = E_2 - E_1$. In both panels we set $M = 2$ and $N = 1$. **a** The space of passive states can be represented in a two-dimensional plot, whose axes represent the probability of occupation of the ground state $p_0$, and first excited state $p_1$. The purple region represents the subset $R_1$, defined in Eq. (23), while the light blue region represents the subset $R_2$, defined in Eq. (24). The black line is $R_3$, that is, the set of completely passive states, see Eq. (25). **b** For a given cycle $S_{m,n}$ we can draw the set of passive states from which energy can be extracted, $R_{m,n}^+$. In this plot, we show the regions $R_{3,1}^+$ (light blue), $R_{5,2}^+$ (orange) and $R_{11,5}^+$ (yellow), which cover the region $R_1$ (purple) better and better as $m$ and $n$ (the number of hot and cold swaps, respectively) grow

divide the set of passive states into three different subsets, namely

$$R_1 = \left\{ \rho_P \text{ passive} \left| \left( \frac{p_1}{p_2} \right)^N > \left( \frac{p_0}{p_1} \right)^M \right. \right\}, \tag{23}$$

$$R_2 = \left\{ \rho_P \text{ passive} \left| \left( \frac{p_1}{p_2} \right)^N < \left( \frac{p_0}{p_1} \right)^M \right. \right\}, \tag{24}$$

$$R_3 = \left\{ \rho_P \text{ passive} \left| \left( \frac{p_1}{p_2} \right)^N = \left( \frac{p_0}{p_1} \right)^M \right. \right\}. \tag{25}$$

The union of these three subsets gives the set of all passive states. In particular, one can verify that the subset $R_3$ contains all the completely passive states, that is, the thermal states of $H_P$ at any temperature $\beta^{-1} \geq 0$. Moreover, $R_1$ corresponds to the set of passive states with $\beta_{\text{hot}}$ associated with the pair of eigenstates $|0\rangle_P$ and $|1\rangle_P$, and $\beta_{\text{cold}}$ associated with the pair $|1\rangle_P$ and $|2\rangle_P$. The set $R_2$, instead, contains the passive states with opposite hot and cold virtual temperatures. Since we are considering qutrit systems, we can represent the set of passive states in a two-dimensional diagram, using their probability distribution. Each point in this diagram represents a passive state. In Fig. 4, we show the three subsets of Eqs. (23), (24) and (25).

In the previous section, we have seen that a cycle defined by the parameters $m$ and $n$ can activate a passive state $\rho_P$ with Hamiltonian $H_P$ if conditions 1 and 2 are satisfied. These conditions apply to the case in which the passive state is described by Eqs. (6) and (7), with $\beta_{\text{hot}} < \beta_{\text{cold}}$. In the present, more general scenario we find that work is extracted by the cycle if and only if

the passive state belongs to the following subset

$$R_{m,n}^+ = \left\{ \rho_P \text{ passive} \mid \left(\frac{p_1}{p_2}\right)^n > \left(\frac{p_0}{p_1}\right)^m \text{ when } m\,\Delta E_{10} - n\,\Delta E_{21} > 0 \right.$$
$$\left. \vee \left(\frac{p_1}{p_2}\right)^n < \left(\frac{p_0}{p_1}\right)^m \text{ when } m\,\Delta E_{10} - n\,\Delta E_{21} < 0 \right\},$$

(26)

where these conditions can be obtained by analysing the general expression of the extracted work. We now show that, by tailoring the value of the parameters $m$ and $n$, we can make $R_{m,n}^+$ to (asymptotically) cover either the region $R_1$ or $R_2$. Here, we focus on $R_1$ solely, since $R_2$ follows from similar arguments. As a first step, we ask $m\,\Delta E_{10} - n\,\Delta E_{21} > 0$, which implies $m > \frac{M}{N}n$, due to Eq. (22). Then, in order to satisfy this condition, we set $m = \frac{M}{N}n + 1$, where we ask $n$ to be large enough for $m$ to be an integer. The set of passive states activated by the cycle is such that

$$\left(\frac{p_1}{p_2}\right)^n > \left(\frac{p_0}{p_1}\right)^m \Rightarrow \left(\frac{p_1}{p_2}\right)^n > \left(\frac{p_0}{p_1}\right)^{\frac{M}{N}n+1} \Rightarrow \left(\frac{p_1}{p_2}\right)^N > \left(\frac{p_0}{p_1}\right)^{M+\frac{N}{n}}.$$

(27)

We notice that, since $\rho_P$ is passive, $p_0 \geq p_1$, which implies that

$$\left(\frac{p_0}{p_1}\right)^{M+\frac{N}{n}} \geq \left(\frac{p_0}{p_1}\right)^M.$$

(28)

Thus, Eq. (27) and (28) together assure that $R_{\frac{M}{N}n+1,n}^+ \subset R_1$. Moreover, if $n \to \infty$, we have that $M + \frac{N}{n} \to M$, which implies that $R_{\frac{M}{N}n+1,n}^+ \to R_1$. Thus, we have that, for a given Hamiltonian $H_P$, and a given passive state $\rho_P \in R_1$, there exist a cycle $S_{m,n}$ such that $\rho_P \in R_{m,n}^+$. However, the closer (in trace norm) the state $\rho_P$ is to the set of completely passive states ($R_3$), the larger the parameters $m$ and $n$ have to be, that is, the larger the machine has to be (Fig. 4).

Work extraction from a generic qudit passive state can be achieved straightforwardly by applying the cycle of the previous section to just three of the $d$ possible levels of the system. The only requirement for the three levels is that their virtual temperatures have to be different. In this case, as we show in more details in Supplementary Note 2, work is extracted from the qudit state by letting the machine interact with the three levels of the system.

**General instability of passive states**. We can now establish our central claim: that any athermal passive state is energetically unstable under a reversible process that does not generate entropy. We analyse the evolution of a passive state, which sequentially interacts with an infinite-dimensional machine $M$, and find that the system moves through a continuous trajectory of passive states towards the set of minimum energy states, that is, the set of the states[45].

We consider a cycle composed of infinitely many hot swaps, $m \to \infty$, and infinite many cold swaps, $n \to \infty$, with the assumption that $n = \alpha m$, where $\alpha$ is a parameter taking values in a specific range, we will describe shortly. Let us now consider the situation in which the main system is a qutrit with Hamiltonian $H_P$ given in Eq. (4), described by the passive state $\rho_P$ whose probability distribution satisfies the equalities of Eqs. (6) and (7). Then, $\rho_P$ belongs to the subset $R_1$ defined in Eq. (23), and the cycle $S_{m,n}$ has to satisfy conditions 1 and 2 in order to extract work from it. These conditions are reflected in the allowed range of the

parameter $\alpha$, that is

$$\frac{\beta_{hot}\Delta E_{10}}{\beta_{cold}\Delta E_{21}} < \alpha < \frac{\Delta E_{10}}{\Delta E_{21}}.$$

(29)

If we set $\alpha$ equal to a value inside the range specified by the previous equation, and we send $m \to \infty$, we find that (see Supplementary Note 4 for details) the state of the machine as obtained from Eq. (9) is given by a mixture of two 'thermal' states, one with effective temperature $\beta_{hot}^{-1}$, the other with effective temperature $\beta_{cold}^{-1}$ (note we still have $H_M = 0$ for the machine). These distributions have support in two different subspaces, and their weight depends non-trivially on the energy gaps of $H_P$ and on the virtual temperatures of $\rho_P$. In fact, we can loosely interpret the state of the machine in terms of a thermal mixture

$$\rho_M = \lambda \tau_{\beta_{hot}} + (1 - \lambda)\tau_{\beta_{cold}},$$

(30)

where

$$\tau_{\beta_{hot}} = \frac{e^{-\beta_{hot}H_{hot}}}{Z_{hot}}, \text{ with } H_{hot} = \sum_{j=0}^{m-1} j\,\Delta E_{10}|j\rangle\langle j|_M \text{ and } Z_{hot} = \text{Tr}\left[e^{-\beta_{hot}H_{hot}}\right],$$

(31)

and

$$\tau_{\beta_{cold}} = \frac{e^{-\beta_{cold}H_{cold}}}{Z_{cold}}, \text{ with } H_{cold}$$
$$= \sum_{j=0}^{n-3} j\,\Delta E_{21}|j+m\rangle\langle j+m|_M \text{ and } Z_{cold} = \text{Tr}\left[e^{-\beta_{cold}H_{cold}}\right].$$

(32)

Notice that in order to define these "thermal states", we have introduced two fictitious Hamiltonians, namely, $H_{hot}$ and $H_{cold}$. These operators are necessary if we want to consider the distribution of the machine as the mixture of two thermal distributions, but they do not enter in any way in the derivation of the extractable work. Indeed, as we specified before, the machine M can have any Hamiltonian (it does not modify the amount of work we extract during the cycle), and we choose to use a trivial one $H_M = 0$, so that the machine acts as a memory. The weight $\lambda$ in the mixture is given by

$$\lambda = \frac{1 - e^{-\beta_{cold}\Delta E_{21}}}{1 - e^{-\beta_{hot}\Delta E_{10} - \beta_{cold}\Delta E_{21}}}.$$

(33)

Thus, during the cycle, the passive state $\rho_P$ first interacts with the "hot reservoir", by performing a sequence of swaps between the pair of states $|0\rangle_P$ and $|1\rangle_P$ and the levels of $\tau_{\beta_{hot}}$. Then, the state interacts with the "cold reservoir", performing a sequence of swaps between the pair $|1\rangle_P$ and $|2\rangle_P$ and the levels of $\tau_{\beta_{cold}}$.

In this scenario, we find that the probability distribution of the passive state $\rho_P$ is infinitesimally modified, and consequently the work extracted is infinitesimally small. In particular, we find that the unit of probability, defined in Eq. (21), tends to 0 with an exponential scaling, $\Delta P \propto e^{-\beta_{hot}m\Delta E_{10}}$ for $m \to \infty$. Let us consider the probability distribution of the final state of the system $\tilde{\rho}_P$. Since the distribution only changes infinitesimally during the cycle, we can recast Eqs. (18), (19) and (20) as a set of differential equations. Thus, we can imagine the situation in which infinite many machines are present, so that we can keep infinitesimally changing the state of the main system. In this case, the evolution

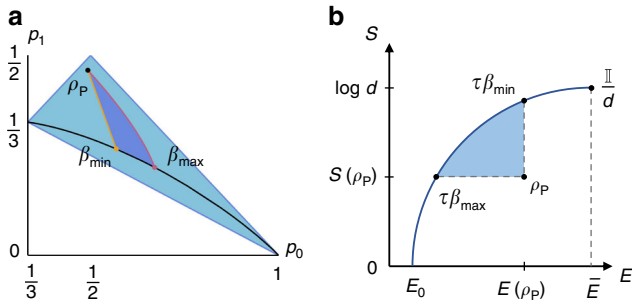

**Fig. 5** Instability of passive states and their dynamics. **a** The state space of a qutrit system, where the region of passive states is highlighted in light blue. The black line contained in the passive region is the set of thermal states. We fix an initial state $\rho_P$, represented by the black point in the diagram. Then, we evolve this state by applying the cycle $S_{m,n}$ (where $m, n \to \infty$) an infinite number of times. The evolution is then modulated by the parameter $\alpha = \frac{n}{m}$. For $\alpha$ equal to $\frac{\Delta E_{10}}{\Delta E_{21}}$, the system evolves along the yellow trajectory, and the final state is the thermal state at temperature $\beta_{\min}$ (with same average energy of $\rho_P$). For $\alpha = \frac{\beta_{\text{hot}}(t)\Delta E_{10}}{\beta_{\text{cold}}(t)\Delta E_{21}}$, the system evolves along the purple line, and the final state is the thermal state at temperature $\beta_{\max}$ (with same entropy of $\rho_P$). The dark blue region represents the subset of achievable states when the initial state is $\rho_P$. **b** A partial representation of the state space of a $d$-level quantum system in the energy–entropy diagram[32]. In this diagram, quantum states are grouped into equivalence classes defined by their average energy $E$ and their entropy $S$. Each point between the $x$-axis (the set of pure states) and the dark blue curve (the set of thermal states) represents one of these equivalence classes. The diagram depends on the Hamiltonian $H_P$ of the system. Here, we only represent the states with average energy lower than $\bar{E} = \text{Tr}[H_P\rho_{\text{mm}}]$, where $\rho_{\text{mm}} = \frac{\mathbb{I}}{d}$ is the maximally-mixed state, since all passive states are contained in this set. For a given initial state $\rho_P$, the light blue region contains all the passive states, which can be achieved with the process

of the state $\rho_P$ is governed by the following equation

$$\frac{dp_1}{dt} = -(1 + \alpha(p_0(t), p_1(t)))\frac{dp_0}{dt}, \qquad (34)$$

where the parameter $t$ provides a continuum label for the sequence of cycles we perform on the passive state. We can then solve this equation for extremal cases for the function $\alpha(p_0, p_1)$. When the parameter function $\alpha(p_0,(t), p_1(t))$ is equal to one of its limiting values, Eq. (34) assumes a clear meaning. In fact,

when $\alpha(p_0(t), p_1(t)) = \frac{\Delta E_{10}}{\Delta E_{21}}$, then the differential equation can be recast as a condition over the average energy of the system, that is,

$$\text{Tr}[H_P\rho_P] = \text{Tr}[H_P\tilde{\rho}_P]. \qquad (35)$$

Then, for $\alpha$ taking this value, the passive state evolves along a trajectory that conserves the energy of the system.

when $\alpha(p_0(t), p_1(t)) = \frac{\beta_{\text{hot}}(t)\Delta E_{10}}{\beta_{\text{cold}}(t)\Delta E_{21}}$, instead, the differential equation can be recast as a condition over the entropy of the system, that is,

$$S(\rho_P) = S(\tilde{\rho}_P), \qquad (36)$$

where $S(\rho) = -\text{Tr}[\rho \log \rho]$ is the Von Neumann entropy. Then, for this $\alpha$, the passive state evolves along a trajectory that conserves the entropy of the system.

For $\alpha$ taking values inside the allowed range, we have that any trajectory between the two presented above is possible, and the set

of achievable states is shown in Fig. 5. It is possible to show that the evolution of the system moves the passive state toward the set of thermal states, which are the stationary states of this dynamic. In Fig. 5, we also show the same set of achievable states, represented this time in the energy-entropy diagram[32]. It is clear that, through this evolution, we can obtain any passive state with a smaller average energy and a bigger entropy than $\rho_P$. In Supplementary Note 5, we show that these states are also the only ones that we can reach with our protocol (and with a broader class of maps, called activation maps).

It is interesting to consider the limiting values of work extraction that can be achieved following the scheme suggested in this section. When the system evolves along the energy-preserving trajectory, the final state we obtain is the thermal state of $H_P$ at temperature $\beta_{\min}^{-1}$, that is, $\tau_{\beta_{\min}}$, where the temperature is such that $\text{Tr}[H_P\rho_P] = \text{Tr}[H_P\tau_{\beta_{\min}}]$. In this case, it is easy to see that the protocol does not extract work, and its only effect consists in raising the entropy of the system. If we consider the efficiency of this cycle, Eq. (17), we see that $\eta = 0$, as expected.

The opposite limit is more interesting. This is the case in which the system evolves along the entropy-preserving trajectory, and the transformation which acts on the system is therefore reversible. The final state we obtain is $\tau_{\beta_{\max}}$, that is, the thermal state of $H_P$ at temperature $\beta_{\max}^{-1}$, such that $S(\rho_P) = S(\tau_{\beta_{\max}})$. The work extracted by the cycle is

$$\Delta W = \text{Tr}[H_P(\rho_P - \tau_{\beta_{\max}})], \qquad (37)$$

which, interestingly, is the maximum amount one can extract by means of reversible operations[31, 32]. We refer to the quantity shown in Eq. (37) as the catalytic ergotropy associated with the passive state $\rho_P$, since it is the maximum energy extracted from the state when reversible operations are allowed in the presence of an additional system (the machine). It is worth noting that the catalytic ergotropy is the optimal work that can be reversibly extracted from a closed system when a catalyst is allowed. In the case of open quantum systems, work extraction aided by auxiliary systems has been considered in refs. [38, 46–48].

Significantly, if we compute the efficiency of the process, Eq. (17), we find that it is equal to the Carnot efficiency, $\eta_{\text{Carnot}} = 1 - \frac{\beta_{\text{hot}}}{\beta_{\text{cold}}}$. Thus, we see that our set up allows for maximal work extraction from passive states, and the protocol extracting this energy is reversible. In fact, no correlations are created between the machine and the system, so that the entropy of the system is preserved. The reversibility of the asymptotic protocol implies (via a Carnot argument[49]) that our result must be optimal, and model-independent. Therefore, even though we extract energy from passive states with a very specific protocol, our result is independent of such protocol, and any other reversible protocol would extract the same amount of energy.

## Discussion

In the paper, we have presented a protocol that allows us to extract work from any single copy of an athermal passive state. The protocol utilises an ancillary system for the work extraction, and the local state of this system is recovered at the end of the cycle. In this way, the cycle can be run multiple times, and each time it acts on a new copy of the passive state. The dimension of the ancillary system grows as the distance between the state and the set of completely passive states is reduced. Moreover, with an infinite dimension machine we can evolve a passive state smoothly towards the set of thermal states. Optimal work extraction can be obtained in this case, and it is achieved by

mapping the initial state into the thermal state with the same entropy.

The present work provides some evidence that a resource theory for thermodynamics with an imperfect thermal reservoir presents non-trivial challenges. Such a resource theory could be realised by providing passive states for free. However, an obvious restriction we should make in this resource theory consists in the fact that we could not provide more than $k-1$ copies of a $k$-activable passive state, otherwise work might be extracted with unitary operations from this free state. Moreover, our results show that, even in the case in which a single passive state is provided, an ancillary system exists such that work can be extracted from the individual passive state. Then, in order to build a sensible resource theory, passive states should be always provided at a work cost, equal to the optimal amount of energy extractable from them when a machine is present.

**Data availability**. Data sharing not applicable to this article as no data sets were generated or analysed during the current study.

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

## Acknowledgements

We are grateful to Ben Schumacher and Michael Westmoreland for inspiring discussions. J.O. thanks the Royal Society and an EPSRC Established Career Fellowship for their support. C.S. is supported by the EPSRC (grant number EP/L015242/1). D.J. is supported by the Royal Society. We thank the COST Network MP1209 in Quantum Thermodynamics.

## Author contributions

All authors contributed to all aspects of this work.

## Additional information

**Competing interests:** The authors declare no competing financial interests.

