## [Peer Review File · Nature Communications]

Reviewers' comments:

Reviewer #1 (Remarks to the Author):

The paper by Sparaciari and coworkers analyses several aspects related to passive states in thermal machines. The paper contains a number of interesting results. There are some points, however, that are unclear and may result in a misleading information conveyed by the work. I would like to ask the authors to consider those points and modify the paper accordingly.

- I find the introduction of the virtual temperatures misleading. As far as I understand the ability to do some work, in the scheme discussed by the authors, is connected to certain relations containing the probabilities and energy differences. I understand that a simple way to parametrising the probabilities is to introduced the virtual temperatures. I fail to understand if there is more than that. The authors seem to convey the impression that the virtual temperatures have some physical meaning.

- Is the swap operation that the authors consider the optimal protocol? I find it natural, I wonder if this can be/ is proved.

- Although I see some obvious differences, several aspects of the protocol discussed in the paper remind Otto machines (see for example New J. Phys. 17 035012 (2015)). May the authors comment on this point?

- On similar grounds, it seems to me that in the section on the instability, the protocols sounds as a generalisation of the homogenisation protocols discussed for example in Phys. Rev. Lett., 88 (2002) 097905.

Reviewer #2 (Remarks to the Author):

The authors consider an abstract and specific mathematical set-up where a heat engine is modelled as a composition of a quantum n -level system and a d -level system with a fully degenerate spectrum. They then consider a specific unitary protocol (made of many swaps) to be performed on this compound and show, in the case of the system being a qutrit, that work can be extracted. They continue with showing that if the initial state is a completely passive state this does not hold true any more. Taking then the limit of infinite number of swaps, they find, depending on a parameter α , various dynamics, that either conserve energy or von-neumann information, or are somewhere in between.

The overall result seems of rather technical nature and their implications for the physics of real system extremely narrow.

A number of remarks are in order.

1) The set-up is very specific on one hand and very far detached from any experimental practical application on the other. In particular a real heat engine is something very different from the set-up studied here, and encompasses at least two heat reservoirs (macroscopic systems with a very large heat capacity) plus a working substance. Identifying the heat bath as a mere pair of energy levels of a single q-dit, is extremely un-physical: in fact physically incorrect.

2) The observation that work can be extracted when unitary operation are allowed onto a larger Hilbert space than that of the passive state system is rather obvious, and also very well known in the literature. Similarly it is rather obvious that this does not happen with a completely passive state: that is in fact what defines a "completely passive state"

3) From the observation of the (known) facts above, restricted here to a special and abstract set-up, the authors draw general claims, such that they so found a “more physically motivated identification of thermal states and emergence of temperature” see abstract. Thermal states and emergence of temperature are well established concepts in statistical mechanics, which have been studied for more than 1.5 centuries under a number of different approaches. The present approach is extremely far from challenging them and to add any further *physical* insight.

For the reasons above I cannot recommend the publication of this paper.

Reviewer #3 (Remarks to the Author):

The authors discuss the concept of passive states in a catalytic setting and show that all extractable work contained in a passive state can still be extracted without access to multiple copies. Passive states are such that no unitary operation can lower their average energy anymore. That means even a controlled cycle of a machine acting on the system with a time-dependent Hamiltonian can extract any amount of work. The authors extend this setting by allowing that unitary to also act on ancillary systems, but with the additional condition that the this ancilla must be returned in the identical state. From a fundamental perspective, this is a very reasonable setting and it was surprising to see that indeed all difference between the average energy to an equi-entropic thermal state is extractable.

This shows, once and for all, that given a sufficient complexity of implementable unitary operations, a non-thermal passive state is indeed a stronger resource than the corresponding thermal state. Such a statement is definitely of a fundamental and universal character and thus a key insight in the field. As the authors correctly point out in their conclusion, the result carries another important implication: All resource theories that take passive states as free states (as opposed to just thermal states in regular thermodynamic resource theories) will ultimately have to include additional limitations on the complexity of the free operations.

The result is mathematically straightforward, easy to understand and well presented with all relevant literature cited. I thus believe that it should be accepted in its current form.

Reviewer #4 (Remarks to the Author):

The present manuscript describes the identification of thermal states and the notion of a temperature from complete passivity. To this end, it is shown that from any athermal quantum state an optimal amount of work can be unitarily extracted in a reversible cycle. This implies that that every passive state except the thermal state is thermodynamically unstable.

The emergence of thermodynamic equilibria in quantum systems is one of the big open questions in quantum thermodynamics. Whereas for classical systems we have a pretty solid understanding, the situation is rather involved in quantum systems. Classical, isolated systems relax into a microcanonical equilibrium state if their dynamics is ergodic and chaotic. Since this notion of equilibration rests on the very classical notion of trajectories, the argument cannot easily be extended to quantum systems. Currently a few very distinct approaches are being developed in the literature, such as the eigenstate thermalization hypothesis (ETH), thermodynamic states from properties and symmetries of entanglement, and resource theoretic approaches. The present manuscript falls into the last class, and presents a more careful, thermodynamic study of thermal, passive states. Within this approach the analysis appears sound, and the result is interesting. However, I cannot recommend the present manuscript to be accepted for publication in nature Communications.

My recommendation is based on the following issue that the authors may find interesting to

address:

In Section I the authors introduce their notion of passive states, which appears to be rather standard. However, some of the definitions and assumptions seem to be physically quite inconsistent. Motivated to get to the ground of this I carefully looked at the cited references [39-41], and unfortunately I failed to resolve the issue: What the authors call a "thermal state" is the quantum equivalent of a Gibbs distribution. From a statistical mechanics point of view this is the stationary distribution of a quantum systems that is in (ultraweak) contact with a thermal reservoir at inverse temperature β . Physically it is important to note that this is the equilibrium state of a thermally OPEN system. Classical thermally isolated systems, as well as quantum systems fulfilling the ETH rather relax into microcanonical states. Similarly all entanglement based approaches also obtain the microcanonical state for isolated systems. The reason why this is important becomes clear when one looks at Eq. (1). Here passivity is defined in terms of unitary operations, which describe thermally isolated dynamics.

So, passivity is defined for thermally isolated dynamics on states that can only be obtained for thermally open systems.

Thus, the present analysis appears only mathematically sound, but the underlying assumptions seem to be rather physically inconsistent.

LETTER TO REFEREES

We would like to thank all the referee for their very helpful comments, and will reply to their concerns. The referees were divided with two positive reports and two negative ones. Referee 3 thought our result was “definitely of a fundamental and universal character and thus a key insight in the field,” and recommended publication. Referee 1 thought “The paper contains a number of interesting results,” but asked us to modify the paper to address some unclear points (which we have done in the resubmission). On the other hand, Referee 4 thought that although “the analysis appears sound, and the result is interesting,” there was a conceptual issue which prevented publication we might find interesting to address. Referee 2 thought the “implications [of our result] for the physics of real system was extremely narrow... A real heat engine is something very different from the set-up studied here, and encompasses at least two heat reservoirs (macroscopic systems with a very large heat capacity) plus a working substance.” We believe we can completely resolve the conceptual issue of Referee 4 and have modified our paper accordingly. We hope to convince Referee 2 of the importance of our result for thermodynamics of small quantum systems. We think in part we over-emphasised the technical aspects of our work, rather than properly explaining the physical question we address. While the physical situation we consider, does not meet the referee’s criteria of a heat engine interacting with two macroscopic heat baths, we hope we can convince them that other situations are also of interest, particularly in light of efforts to build microscopic thermal machine interacting with single small reservoirs in the quantum regime.

Finally, in light of the scepticism of Referee 2, we would like to emphasis the surprising nature of our result. Indeed, there is an important paper titled “Maximal work extraction from quantum systems”, *EPL (Europhysics Letters) 67, 4 (2004)* which has over 75 citations in google scholar. Our work shows that even the title of that paper leads one astray, since it implies that one cannot extract work from a passive state in any cyclic process (where *ancillary systems* and machines are returned back to their original state).

We respond to each referee below.

Sincerely,
David Jennings, Jonathan Oppenheim, Carlo Sparaciari

Referee 1:

The referee lists a number of points which need clarification and says “I would like to ask the authors to consider those points and modify the paper accordingly.” We believe we can do so, and describe this below:

- *“I find the introduction of the virtual temperatures misleading.”*

We agree that this is perhaps a confusing use in the context. We have thus clarified in the main text that virtual temperature is just a convenient way of parametrise the passive states. We believe this parametrisation to be useful (in this context) since the catalyst that we construct acts as a kind of quantum memory, and the protocol can be visualised as the cycle performed by a machine interacting with reservoirs with “virtual temperatures”. However, at all times the Hamiltonian of the catalyst is trivial and there is no actual thermality present in either system. Thus, these virtual temperatures do not have the same physical meaning as traditional temperatures, but do have physical consequences so as to determine the dynamics that the unitary protocol generates.

- *“Is the swap operation that the authors consider the optimal protocol? I find it natural, I wonder if this can be/ is proved.”*

This is an interesting question and indeed we can prove the protocol is optimal. This can be shown since the asymptotic protocol is reversible (see end of Section I.D., and compare it with the result of Ref. [29]). Our result is therefore a model-independent statement. The existing protocol is quite complex, and we suspect a simpler formulation could be found. Indeed we understand that the Bristol group, who were very surprised and excited by our result, are working on finding a simpler protocol. We have added a comment to this effect in the main section. Namely we extended the last paragraph of section I.D., in which we originally pointed out that our protocol can achieve optimal work extraction.

- *“Although I see some obvious differences, several aspects of the protocol discussed in the paper remind Otto machines (see for example New J. Phys. 17 035012 (2015)). May the authors comment on this point?”*

Our protocol is inspired by the cycles performed by heat engines. It should not be surprising, then, to find some similarity between our protocol and such cycles. Furthermore, the scheme of the Otto engine seemed particularly appropriate for our problem (since, for instance, a protocol shaped over the Carnot cycle would present several difficulties in this scenario, not least the need of performing isothermal transformation in the absence of real thermal reservoirs). We have included the reference – New J. Phys. 17, 035012 (2015) – for completeness.

- *“On similar grounds, it seems to me that in the section on the instability, the protocols sounds as a generalisation of the homogenisation protocols discussed for example in Phys. Rev. Lett., 88 (2002) 097905.”*

The protocol we utilise to prove the instability of passive state does share some conceptual features with the one suggested by the referee. However, we would like to stress that the homogenization protocol provided in the paper explicitly refers to the presence of an external reservoir (see also the abstract of [arXiv:quant-ph/0110164](https://arxiv.org/abs/quant-ph/0110164), where the protocol was first introduced), and to the notion of equilibration. In our case, instead, we are dealing with closed system dynamics in the presence of catalysts. Therefore, while conceptually the two protocols might share some similarity, physically they refer to two different scenarios. We have included the reference – *Phys. Rev. Lett.* **88**, [097905 \(2002\)](https://doi.org/10.1103/PhysRevLett.88.097905) – for completeness.

Referee 2:

We thank the referee for their comments, and would like to address some of the points that were raised. In particular, we would like to elaborate more on the core setting and aims of the work, since the existing paper perhaps did not convey them in a sufficient manner. We also indicate the changes we have made to address their concerns.

Firstly, it is important to emphasize that in the paper we neither consider a heat engine nor a statement about thermalisation processes. We also agree with the referee that it is entirely obvious that work can be extracted from a passive, but not completely passive, state via extension to a larger Hilbert space, and if this were the statement of our work we would completely concur with the referee. Indeed, extension to a larger Hilbert space involves either assuming unoccupied levels (in which case the total state is not passive) or using another system which need not be returned to its original state (and therefore it is trivial to extract work). We consider neither of these, which would constitute a form of “cheating” the second law.

However, the aim of our work is not to construct another heat engine, or to show thermalisation of systems. The core question, instead, is the following: given a closed system consisting of a reservoir in some state and a machine that interacts with it, which states of the reservoir allow the machine to extract work **in a cyclic process**. The challenge is to know whether one can *in principle* construct a protocol involving closed system dynamics, which leaves the auxiliary system precisely unchanged at the end. This is a very real and physically motivated question. Given the importance of thermodynamics as a field, surely the answer to such a fundamental question is not narrow?

In the thermodynamic limit, the answer to our core question is obviously any state but the thermal state. We totally agree with the referee that this has been known “for more than 1.5 centuries under a number of different approaches.” However, for small reservoirs, it was believed that the answer to this question was any non-passive state. We know of no previous claim that one can extract work from a single passive state. We only know of claims that one can extract work in the un-physical situation where one has a large number of identically prepared passive states. Indeed this is all that people do. However, crucially, this is not a physically motivated scenario when one is not operating in the thermodynamic limit. Understanding thermodynamics when one does not take the thermodynamic limit is a central focus of a lot current work.

Our result is thus surprising – you can in principle extract work from a single non-thermal passive state. It is true that extracting such work from passive states may be difficult in practice, but the fact that you can do it is surely of fundamental importance. And especially important if we are to understand thermodynamics at small scales, where such states are ubiquitous. We have modified the introduction of the paper to make clearer in which sense (and for which scenario) our results singles out the thermal states.

The referee says that “The set-up is very specific on one hand and very far detached from any experimental practical application on the other. In particular a real heat engine is something very different from the set-up studied here, and encompasses at least two heat reservoirs (macroscopic systems with a very large heat capacity) plus a working substance. Identifying the heat bath as a mere pair of energy levels of a single q-bit, is extremely un-physical: in fact physically incorrect.” Regarding practical applications, we would again emphasize that the aim of the paper is expressly not the construction of heat engines for practical settings, and we apologise if the writing of the paper suggested this. Rather we address the question of whether it is possible in principle to extract work from passive states. We have now modified the introduction in order to make this concept clearer.

Regarding the specific nature of our protocol, it is true that we use a very specific set-up, however this does not imply that the conclusion is restricted or specialised in any sense to this particular model. This follows because the asymptotic protocol we construct generates zero entropy through correlations, and is therefore asymptotically reversible. Since it is asymptotically reversible it follows (via a Carnot argument) that it must be an optimal, model-independent statement. It may be possible to construct a simpler protocol that obeys the crucial requirements, but this appears to be a non-trivial task. Irrespective of such a simplification, the asymptotic reversibility makes the model use irrelevant to the central aim of our work. One might be concerned that we allow unitaries in our protocol, but it has been well established that this can be converted to an autonomous machine, or to other paradigms typically considered in thermodynamics – see appendix H of *Phys. Rev. Lett.* **111**, 250404 (2013).

Regarding the comment that a real heat engine must operate between two heat baths and that these must be macroscopic, we think this unnecessarily rules out a huge number of physically interesting situations. To refuse to consider a microscopic heat engine which interacts with a part of a single reservoir is to ignore much of quantum thermodynamics. Considering new situations is what will allow the field of thermodynamics to evolve and gain new insights. We have modified our paper to use the terminology of *thermal machine*, rather than heat engine, in order to make a distinction with the classical, macroscopic case.

As stated our work is not a thermalisation analysis. Instead it provides a rigorous statement on passivity in individual quantum systems. It is of increasing interest to make statements that apply to individual quantum systems, instead of results that require multi-partite interactions that either extract work in some way, or thermalise through the generation of correlations. The traditional accounts of going from passivity to complete passivity, classical thermalisation theory, or eigenstate thermalisation hypothesis scenarios are all of the latter kind. In contrast, our statement can be made deterministically for a single closed quantum system.

Referee 4:

The referee found our results interesting and technically correct, but felt that our paper should be rejected because of a conceptual issue. Namely, that the thermal state is derived in the context of open systems, while we consider unitary operations (closed system dynamics). Namely, they write:

“So, passivity is defined for thermally isolated dynamics on states that can only be obtained for thermally open systems. Thus, the present analysis appears only mathematically sound, but the underlying assumptions seem to be rather physically inconsistent.”

We thank the referee for their very useful comments and insights, and would like to expand on the context and assumptions of our work some more. We believe we can modify the paper to completely resolve the conceptual issue they point out. Namely, we can view any passive state as arising as part of a pure quantum state, and it is simply this component on which the protocol is performed. In other words, we can, as the referee demands, only consider closed system dynamics. We have modified our paper so that we are now consistent as the referee demands, and now emphasize how our question arises for closed systems. Namely, consider the micro-canonical state (or pure state in the quantum case) as the referee suggests we consider. Then one can ask, given access to part of this state (or some degrees of freedom, usually local), what state on this subsystem allows one to extract work from it, i.e. is unstable. Previously, one would have thought that if the subsystem is in a passive state, then it is stable and no work can be extracted. We show for the first time that this is incorrect.

Since this was their only objection, and they did find our paper interesting, we hope this is able to satisfy them as to the correctness of our work.

We should also emphasize that our work is not aiming to make statements within the topic of equilibration theory, or thermalisation, and while passive states naturally emerge in thermalisation scenarios, we would argue that it is a distinct focus. The statements of passivity and complete passivity can be viewed as statements of energetic stability (in the purely mechanical sense) for a state in a closed system undergoing unitary dynamics. We fully agree that such passive states naturally arise within a range of thermalisation settings (as well as settings involving non-equilibrium steady states) in which a multi-partite system (either in a pure or mixed state) undergoes dynamics that typically produces thermalisation on the margins of sufficiently small systems.

Our work is in a different context. The central question is: does (non-thermal) passivity within a *single* closed system exist in the context of catalysts? Firstly, this is a statement about an individual quantum system, and secondly it prohibits an answer in terms of thermalisation, in the sense described by the referee. Crucially no entropy can be generated through the build-up of correlations, and the single catalyst state must finish exactly as it started. Our result should be interpreted as a single-shot statement about Gibbs states of quantum systems, independent of thermalisation. Prior analyses, such

as the transition from passivity to complete passivity in the IID limit, thermalisation theory, and typicality in entangled pure states, are crucially multi-partite, require probabilistic statements, and require the build-up of correlations. The argument we present here is a deterministic, single-shot protocol in which a perfect catalyst is constructed and yields a reversible protocol in which zero correlations/entropy is generated. This is important because in any physical setting, one has a single reservoir, not many copies of that reservoir.

Finally, we would like to stress that, while our result does not concern thermalisation of quantum systems, it brings new ideas into the study of a fundamental concept in thermodynamics, namely, passivity (introduced in two seminal papers – by Pusz, Woronowicz and by Lenard – that nowadays count more than 400 citations in google scholar). Furthermore, this result concerns thermodynamics of individual quantum systems, a topic of increasing interest within the quantum thermodynamics community (see, for instance, Nat. Commun. 5:4185 (2014), and arXiv:1702.08473).

CHANGES TO THE MAIN PAPER

- Clarified the nature and use of virtual temperatures in our work:
 - paragraph starting at line 24
 - paragraph starting at line 96
 - paragraph starting at line 101
- Discussion on the optimality of the protocol:
 - paragraph starting at line 265
- Added the references suggested by the referee:
 - reference 27 and 28
- Clarified how the thermal state is singled out in our framework:
 - paragraph starting at line 48
- Clarified the main objectives and framework:
 - paragraph starting at line 15
 - paragraph starting at line 24
 - paragraph starting at line 33
 - paragraph starting at line 41
- Used the term “thermal machine” instead of “heat engine” to avoid confusion with the macroscopic case.
- Add the concept of catalytic ergotropy and a brief discussion of it:
 - paragraph starting at line 259

REVIEWERS' COMMENTS:

Reviewer #1 (Remarks to the Author):

I am ok with the reply of the authors and the revised version of the paper. No further comments on my side. the paper may be accepted in Nat. Comm.

Reviewer #2 (Remarks to the Author):

The authors have responded to the criticisms raised and have made minor changes to the paper.

I remain convinced that the paper presents a technical result of neither fundamental nor experimental relevance. Surely the paper does not have the broad implications claimed by the authors, while being potentially misleading to new researchers entering the field of statistical mechanics and thermodynamics of small quantum systems, by bringing up and mixing a number of issues (e.g. regarding thermalisation, heat engine operation, the concept of temperature, ergotropy) with the result of confusing rather than clarifying.

I believe this paper does not enjoy that broadness of views and sharpness of message that is necessary for a paper to advance the fundamental understanding of the thermodynamics of small quantum systems. Hence I remain with the opinion that this paper is not suitable for publication in Nature Communications.

Reviewer #4 (Remarks to the Author):

In my previous report on the manuscript I was not able to recommend publication. Although I was convinced that the research is mathematically sound and interesting I had a conceptual issue with the interpretation of the physics. In their reply and in their revisions the authors have thoroughly and convincingly address my concerns. To be honest, resolving the issue with the help of purification is rather neat and I am eager to think more about the consequences. Thus, I can now recommend the manuscript to be accepted for publication.

Reply To Referees

We again thank the referees for their extremely helpful comments and provide responses to each below.

Best wishes,
The Authors.

Referee #1:

No further issues were raised and we thank the referee again.

Reviewer #2:

"The authors have responded to the criticisms raised and have made minor changes to the paper. I remain convinced that the paper presents a technical result of neither fundamental nor experimental relevance. Surely the paper does not have the broad implications claimed by the authors, while being potentially misleading to new researchers entering the field of statistical mechanics and thermodynamics of small quantum systems, by bringing up and mixing a number of issues (e.g. regarding thermalisation, heat engine operation, the concept of temperature, ergotropy) with the result of confusing rather than clarifying. I believe this paper does not enjoy that broadness of views and sharpness of message that is necessary for a paper to advance the fundamental understanding of the thermodynamics of small quantum systems. Hence I remain with the opinion that this paper is not suitable for publication in Nature Communications."

Reply to Reviewer #2:

We are sorry to see that our reply was not sufficient to convince the Referee of the fundamental nature of our result. However, we believe that, thanks to the comments received in the initial review process, the paper is now clearer and accessible to a broad audience, including new researchers entering this field.

Referee #3:

No further issues were raised and we thank the referee again.

Referee #4:

No further issues were raised and we thank the referee again.